# Three-Dimensional Printing Quality Inspection Based on Transfer Learning with Convolutional Neural Networks

**DOI:** 10.3390/s23010491

**Published:** 2023-01-02

**Authors:** Cheng-Jung Yang, Wei-Kai Huang, Keng-Pei Lin

**Affiliations:** 1Program in Interdisciplinary Studies, National Sun Yat-sen University, Kaohsiung 80424, Taiwan; 2Department of Information Management, National Sun Yat-sen University, Kaohsiung 80424, Taiwan

**Keywords:** fused deposition modeling, image analysis, quality inspection, transfer learning, ensemble learning

## Abstract

Fused deposition modeling (FDM) is a form of additive manufacturing where three-dimensional (3D) models are created by depositing melted thermoplastic polymer filaments in layers. Although FDM is a mature process, defects can occur during printing. Therefore, an image-based quality inspection method for 3D-printed objects of varying geometries was developed in this study. Transfer learning with pretrained models, which were used as feature extractors, was combined with ensemble learning, and the resulting model combinations were used to inspect the quality of FDM-printed objects. Model combinations with VGG16 and VGG19 had the highest accuracy in most situations. Furthermore, the classification accuracies of these model combinations were not significantly affected by differences in color. In summary, the combination of transfer learning with ensemble learning is an effective method for inspecting the quality of 3D-printed objects. It reduces time and material wastage and improves 3D printing quality.

## 1. Introduction

The advent of Industry 4.0 has birthed a fresh pursuit for increasingly productive and cost-efficient manufacturing technologies, and three-dimensional (3D) printing has become a key technology for parts manufacturing. Fused deposition modeling (FDM) 3D printing is currently the most popular type of 3D printing in the consumer space, and it has found applications in many industries [1,2,3,4]. FDM can be used to quickly generate proofs of concept for geometrically complex products [5,6], and it is a valuable tool for versatile manufacturing because of its ability to use a wide variety of filament materials [7]. Furthermore, FDM requires little postprocessing and has short processing times [8,9,10,11]. However, it takes many hours to produce large parts via FDM printing, and it is possible for various defects to form during this process, which may degrade the final product or cause it to fail. This increases the time, material, and effort required. Therefore, it is necessary to monitor printing quality, and visual inspection is by far the best way to obtain timely feedback. Hence, many technological innovations for this purpose have been based on visual inspection.

There are numerous examples in the literature where machine learning and deep learning have been successfully used to detect warping and delamination, measure surface roughness, monitor the 3D printing process, and predict printing quality. For example, supervised learning algorithms such as the naive Bayes classifier [12,13,14], k-nearest neighbors [13], random forest [13], decision tree [13,14], and support vector machine [13,15,16] have been used to train models to predict and detect defects. Accuracy comparisons have been performed among various convolutional neural networks (CNNs) [13,17,18,19]. Furthermore, CNNs have been combined with various machine learning methods to evaluate the accuracy of defect-detection models [20,21,22]. The CNNs have also been utilized in image quality assessments [23]. Zhou et al. proposed to utilize a dual-stream convolutional neural network that simulates dual views of the human visual system to predict the perceptual quality of stereoscopic images [23].

Over time, numerous CNNs have been developed to increase the classification accuracy. Simonyan and Zisserman were the first to propose the use of very deep CNNs to increase the fineness of the extracted features [24], which culminated in the emergence of the Visual Geometry Group (VGG) series of CNNs for image recognition and classification [25,26,27,28]. Szegedy et al. proposed the Inception architecture, which increases the computational efficiency and facilitates increases in the network depth and width [29]. Szegedy et al. subsequently added batch normalization layers to regularize intermediate features, which significantly accelerate learning and convergence, and they reduced the size of the feature map by using factorization with parallel pooling and convolution [30]. Since the gradients may vanish or explode in very deep networks, and degradation can occur if the depth of an optimal shallow network is increased, He et al. proposed the learning of residuals to address this problem. By adding residual connections (skip connections) to the outputs of a CNN, the accuracy could be improved in very deep networks. This led to the creation of the ResNet series [31,32]. ResNet has been used for fault diagnosis in industrial manufacturing [33,34], rolling bearings [35], and rotating machinery [36]. The increases in network depth and width facilitated by the ResNet architecture led to tremendous improvements in CNN performance. ResNet has also been utilized as a pretrained model for image quality assessment. Sun et al. utilized six ResNet34s to extract features from six views of 360-degree images and combined them with a regressor for image quality assessment of VR devices [37].

In 2019, Tan et al. used the neural architecture search (NAS) technique in Google AutoML to design a baseline network, and they scaled the width, depth, and resolution of this network using the compound coefficient to create a family of models called EfficientNet [38]. In 2021, Tan et al. published the v2 versions of the EfficientNet models, which are smaller and faster to train than their predecessors [39]. Zhou et al. used EfficientNetV2 for fine-tuning the EfficientNet-B7 pretrained model to create a machine that extracts features from whole-body portraits and generates Lego brick models [40]. This approach was also used for dog nose print matching [41] and prediction of the energy consumption of 3D printing processes [42].

Kim et al. adopted VGG19 pretrained for transfer learning to detect the filament tangling in the 3D printing process [28]. Baumgartl et al. used thermographic imaging data to train a CNN to detect defects in laser powder bed fusion in metal 3D printing [18]. Banadaki et al. used Inception-v3, which is also based on a convolutional neural network, for real-time surface defect detection and grading during FDM printing [43]. Jin et al. proposed the use of Inception-v3 as a transfer learning model to classify the extent of delamination and predict warping [33]. Razaviarab et al. proposed the use of transfer learning in combination with a closed-loop machine learning algorithm to automatically detect defects in 3D printing [44].

The work of Kadam et al. [13] is most relevant to our work, which studied the fault detection of FDM by combining the pretrained models (AlexNet, GoogLeNet, ResNet18, ResNet50, EfficientNet-B0) with popular classification methods including KNN, SVM, naïve Bayes classifier, etc. Their results showed that the SVM achieved the best accuracy in four of the five combined pretrained models, and the combination of SVM with AlexNet resulted in the best accuracy.

Combining CNN with different machine learning methods usually results in good performance for fault detection and classification. Although ensemble learning is an important part of machine learning, the combination of transfer learning and ensemble learning for surface defect detection in 3D printing, which would reduce time and material wastage, has yet to be investigated. Furthermore, although color selection is critical for FDM-printed objects, there are no reports about the effects of color on the accuracy of defect-detection algorithms. Therefore, CNN-based transfer learning was combined with six ensemble learning models to detect and classify surface defects in FDM-printed 3D objects. The effects of color on the accuracies of these model combinations were analyzed. The findings of this study reveal the model combinations that are most accurate for each color and geometry.

There have been reports in bearing fault diagnosis that using CNN for feature extraction with the boosting method LightGBM for classificiation can result in better accuracy than using only CNN [45]. The transfer learning with CNN-based pretrained models on large-scale image datasets such as GoogleNet and AlexNet has been successfully utilized on 3D printing image classification [13]. In this work, we further combine the transfer learning with bagging and boosting approaches to make an ensemble approach.

The neural network models usually have high variances, and in 3D printing image classification, the dataset is usually rather small, which may increase the variance of prediction. Therefore, we capitalize on the ensemble approach, the bagging and boosting, to improve the performance on training classifiers from the features extracted by the neural networks-based pretrained models.

The bagging algorithm trains a group of classifiers on different subsamples of the dataset to make an ensemble classifier. Bagging can help improve the stability of prediction, especially for unstable approaches such as neural networks. It can also reduce the variances and prevent overfitting, benefitting the 3D printing image classification of small samples, since collecting the samples of 3D printing is very time-consuming, and small samples are prone to overfitting. The boosting algorithm trains a series of weak learners from the extracted features and combine to an ensemble classifier, which is expected to reduce the variances and biases in the classifier, and therefore improve the classification performance.

In addition to combining the existing pretrained models with the ensemble approach, we also study the application of the newly developed pretrained models for transfer learning in 3D printing classification. The newer pretrained models, either featured with deeper layers or different network structure, have been continuously developed to improve the classification performance. The experimental results show that the ensemble approaches, either bagging or boosting, can effectively improve the classification performance. Therefore, by utilizing the transfer learning with newly developed pretrained models and combining them with the ensemble approach, we can achieve a better classification performance in 3D-printed image classification.

The remainder of this paper is organized as follows. Section 2 introduces the materials and algorithms used in this study. Section 3 presents the experimental data and the findings of the data analysis. Section 4 discusses how the printing geometry and filament color are related to the accuracy of each algorithm, as well as the contributions of these factors to algorithm accuracy. Finally, the conclusions and outlook are presented in Section 5.

## 2. Materials and Methods

All the 3D printing was performed using a Prusa i3 mk3s 3D printer (Prusa, Prague, Czech Republic) [46]. with a polylactic acid filament [47]. All the photographs were captured using a Sony a7 III camera (Sony, New York, NY, USA) [48]. Image preprocessing and model training and testing were performed on the Google Colab platform [49].

### 2.1. Classification Principles

To construct an image dataset of defective and nondefective samples based on the classification principles shown in Figure 1, the images were manually labeled one by one. Nondefective samples had smooth and fully filled surfaces; all other samples were classified as defective. The filament colors were gray, green, and blue. Figure 2 shows the pictures of the finished layer. The collected dataset contains the pictures of every printing layer.

### 2.2. Experimental Procedures

The experimental procedures of this study are shown in Figure 3. First, the 3D printer was configured, and a camera was set up to capture photographs of the 3D-printed object. In Step 2, the collected photographs were cropped and classified to create the image dataset. In Step 3, the image dataset was divided into training and testing sets at a 7:3 ratio using split_train_test. In Step 4, the Python Open Source Computer Vision Library was used to extract the red, green, and blue color model (RGB); GRAY; and hue, saturation, and value color model (HSV) values of each image, and the angle of each image was varied to produce images with different angles. In Step 5, a variety of CNNs were used for feature extraction from the images. In Step 6, a variety of ensemble learning algorithms were used to train defect-detection models; the accuracy of each model was then evaluated. Finally, the model accuracies were analyzed. The flow graph of the proposed method is shown in Figure 4.

### 2.3. CNN

The CNNs used in this study were only used as pretrained models for feature extraction from the image dataset. Therefore, a variety of ensemble learning algorithms were used for classification, training, and testing to increase the accuracy. The CNNs used in this study included VGG16, VGG19, InceptionV3, ResNet50, EfficientNetB0, and EfficientNetV2L. A brief overview of these models is presented below.

#### 2.3.1. VGG16

VGG16 is a 16-layer CNN with 13 convolutional layers and 3 fully connected layers [29]. As this network architecture has a large number of weights and fully connected nodes, its parameter space is large, which results in long training times. The size of the VGG16 model is 528 MB, and its input images are 224 × 224 RGB images. The architecture of VGG16 is shown in Figure 5.

#### 2.3.2. VGG19

VGG19 is a 19-layer CNN with 16 convolutional layers and 3 fully connected layers (three more convolutional layers than VGG16) [29]. Similar to VGG16, VGG19 has a large number of weights and fully connected nodes. Its size is 549 MB, and its inputs are also 224 × 224 RGB images. The VGG19 architecture is shown in Figure 6.

#### 2.3.3. InceptionV3

InceptionV3 is the third generation of the Inception architecture. It consists of several block modules, which have a global average pooling layer instead of a fully connected layer at the end. These block modules give InceptionV3 a total of 47 layers [30]. In contrast to ResNet, InceptionV3 avoids representational bottlenecks early in the network to prevent losses of feature information. As the InceptionV3 model can increase the width and depth while maintaining computational efficiency, its size is only 92 MB. Its inputs are 299 × 299 RGB images. The InceptionV3 architecture is shown in Figure 7.

#### 2.3.4. ResNet50

ResNet50 is a part of the ResNet family of models, and as its name suggests, it has 50 layers, which consist of 49 convolutional layers and 1 fully connected layer [31]. As the model uses residual connections instead of fully connected layers and consists of residual blocks, its size is only 98 MB. The inputs of ResNet50 are 224 × 224 RGB images. The ResNet50 architecture is shown in Figure 8.

#### 2.3.5. EfficientNetB0

EfficientNetB0 is a model from the EfficientNet series. It consists of 5 large modules with different numbers of submodules, giving it a total of 237 layers [38]. The MBConv6 modules that constitute EfficientNetB0 are depthwise separable convolutions from the MobileNet architecture with ResNet-like residual connections, and there is an expansion layer that increases the number of channels by a factor of 6 [50]. Furthermore, NAS was used to determine the depth, width, and number of channels of EfficientNetB0. B0 is the smallest model in the EfficientNet series, and its size is only 29 MB. The inputs of this model are 224 × 224 RGB images. The EfficientNetB0 architecture is shown in Figure 9.

#### 2.3.6. EfficientNetV2L

EfficientNetV2L comes from the EfficientNetV2 family of models. To increase the training speed and parameter efficiency, the V2 edition of EfficientNet uses training-aware NAS and scaling and Fused-MBConv modules in shallow networks. EfficientNetV2 also uses an improved progressive learning method, where the regularization strength (dropout rate, RandAugment magnitude, and mixup ratio) is adjusted according to the image size, which increases the accuracy and training speed [39]. As the V2L model contains a large number of layers, its size is 479 MB. Its inputs are 480 × 480 RGB images. Details regarding the architecture of EfficientNetV2L are presented in Table 1.

### 2.4. Ensemble Learning Algorithms

#### 2.4.1. Bagging

The idea of bagging is to randomly sample the training set to train multiple independent classifiers with normalized weights, which then vote on the final result [51,52,53]. Random forest (RF) bagging was used in this study, which is a supervised algorithm. It is an advanced version of the decision tree architecture. An RF consists of multiple decision trees, and besides selecting random samples from the training set (bagging), random subsets of features are drawn to train each tree. Although overfitting tends to occur when a decision tree becomes too deep, the RF architecture resists overfitting by having multiple decision trees, which allows RF to work accurately and efficiently on large high-dimensional datasets [51,52].

#### 2.4.2. Boosting

The idea of boosting is to combine multiple weak classifiers into a single strong classifier. Boosting is an iterative approach, where input data that were misclassified by the older classifier are given a higher weight when training a new classifier. This allows the new classifier to learn features, and thus increases the accuracy. Finally, the iteratively trained weak classifiers vote (with weights) to produce the final result [53,54,55]. The boosting models used in this method were AdaBoost, GBDT, XGBoost, LightGBM, and CatBoost.

1.AdaBoostThe idea of AdaBoost is to create a strong classifier by summing weighted predictions from a set of weak classifiers. AdaBoost, which is short for adaptive boosting, uses the misclassified samples of the preceding classifiers to train the next generation of classifiers. This is an iterative approach where weighted training data are used instead of random training samples, so that the classifier can focus on hard-to-classify training data. A new classifier is added at each iteration, until the error falls below a threshold. As the model is effectively a strong classifier, it is robust against overfitting. However, noisy data and outliers should be avoided to the greatest extent possible [56,57].

2.Gradient-Boosting Decision Tree (GBDT)The GDBT is a multiple-additive regression tree and is a technique where a strong classifier is formed by combining many weak classifiers. The GDBT model is applicable to both classification and regression problems. Every prediction differs from the actual value by a residual; in GDBT, the log-likelihood loss function is used to maximize the probability that the predicted value is the real value. To prevent overfitting, the residual and predicted residual are calculated, and the predicted residual is multiplied by the learning rate. New trees are generated one after another to correct the residual until it approaches 0, that is, until the prediction approaches the true value [58,59,60,61].

3.Extreme Gradient Boosting (XGBoost)XGBoost is a method where additive training is combined with gradient boosting. In each iteration, the original model is left unchanged, and a new function is added to correct the error of the previous tree. The risk of overfitting is minimized through regularization and the addition of a penalty term Ω to the loss function. XGBoost combines the advantages of bagging and boosting, as it allows the trees to remain correlated with each other while utilizing random feature sampling. In contrast to other machine learning methods that cannot handle sparse data, XGBoost can efficiently handle sparse data through sparsity-aware split finding. In this method, the gains obtained from adding sparse data to the left and right sides of a tree are calculated, and the side that gives the highest gain is selected [61,62,63,64].

4.LightGBMLightGBM is a type of GDBT that uses histogram-based decision trees, which traverse the dataset and select optimal splitting points based on discrete values in a histogram. This reduces the complexity of tree node splitting and makes LightGBM very memory- and time-efficient. LightGBM uses gradient-based one-side sampling to retain training instances with large gradients, as well as exclusive feature bundling to reduce the dimensionality [61,65,66,67].

5.CatBoostCatBoost is another GDBT-based model. To create unbiased predictions, CatBoost uses ordered boosting to reduce the degree of overfitting and uses oblivious trees as base predictors. In many competitions hosted by Kaggle, CatBoost achieved the highest accuracies and smallest log-loss values [61,66,67,68,69,70].

## 3. Experiments

The gray dataset contains 1275 samples, where 575 are nondefective and 700 are defective. The green dataset contains 1464 samples, where 735 are nondefective and 729 are defective. The blue dataset contains 1274 samples, where 644 are nondefective and 630 are defective. Detailed statistics for each shape are shown in Table 2. The printing temperature of extruder is 210 °C and bed is 60 °C, and the fill density is 20%. The retraction speed is 35 mm/s, and printing speed is 20 mm/s. The layer thickness is 0.1 mm, and there are two top solid layers.

### 3.1. Effects of Geometric Differences

To present all the pretrained model + ensemble learning combinations and their accuracies, we provide a visual representation of their accuracies on gray-colored geometries in Figure 10. We also show the accuracy of using AlexNet with SVM (raw data are presented in Appendix D), which was the best combination reported in [13], as the baseline for comparison. The performance of the baseline is shown as a dotted regular hexagon in each color–shape combination.

Here, we summarize the results for each gray-colored geometry. Gray squares: For this dataset, VGG16, VGG19, and ResNet50 consistently obtained high accuracies with all the ensemble learning models; EfficientNetV2L, by contrast, performed poorly with all the ensemble learning models. Gray circles: The highest accuracy was achieved by VGG19 + AdaBoost (90.91%), followed by InceptionV3 + AdaBoost (88.64%), whereas EfficientNetB0 had the lowest accuracy with all the ensemble learning models. Gray diamonds: InceptionV3 + CatBoost had the highest accuracy (93.18%), followed by VGG19 + AdaBoost (90.91%). Again, EfficientNetB0 had the lowest accuracy, regardless of which ensemble learning model it was paired with. Gray ovals: The VGG16 + Catboost, VGG16 + Xgboost, VGG16 + GradientBoosting, and ResNet50 + XGboost pairings were tied for the highest accuracy (92.37%), followed by VGG19 + RandomForest and EfficientNetV2L + RandomForest (91.60%). Gray stars: VGG16 + RandomForest, VGG19 + RandomForest, and VGG19 + AdaBoost were tied for the highest accuracy (92.45%); the lowest accuracies were obtained with EfficientNetV2L + any ensemble learning model. Gray triangles: VGG16 + RandomForest achieved the highest accuracy (94.44%), followed by VGG16 + Catboost (92.59%). All gray pictures: VGG16 + LightGBM and VGG16 + GradientBoosting achieved the highest overall accuracy (92.44%), followed by VGG16 + XGboost (91.41%).

### 3.2. Effects of Color

Next, defect classification was performed on the green and blue geometries. Figure 11 shows the accuracy of each model combination for green geometries. Green squares: VGG16 + ResNet50 and VGG19 + ResNet50 were both highly accurate, but in contrast to the case of gray squares, InceptionV3 + RandomForest and EfficientNetV2L + AdaBoost also achieved the highest level of accuracy. Green circles: VGG19 + InceptionV3 and VGG19 + RandomForest were the most accurate combinations (93.94%), followed by VGG19 + LightGBM (92.42%). EfficientNetV2L consistently exhibited the lowest accuracy, regardless of the ensemble learning model. Green diamonds: EfficientNetB0 + LightGBM was the most accurate combination (92.59%), followed by EfficientNetB0 + GradientBoosting (90.74%). Green ovals: VGG19 + CatBoost, VGG19 + InceptionV3, and VGG19 + XGboost were tied for the highest accuracy (98.31%). EfficientNetB0 and EfficientNetV2L were consistently the least accurate models, with all the ensemble learning models. Green stars: ResNet50 + AdaBoost, ResNet50 + GradientBoosting, and EfficientNetB0 + GradientBoosting were tied for the highest accuracy (95.83%), whereas EfficientNetV2L had the lowest accuracy with all the ensemble learning models. Green triangles: VGG16 + LightGBM, VGG16 + AdaBoost, and InceptionV3 + AdaBoost were tied for the highest accuracy (94%), followed by VGG16 + XGboost and VGG16 + GradientBoosting (92%). All green geometries: VGG19 + XGboost and InceptionV3 + Catboost were tied for the highest accuracy (92.33%), followed by VGG16 + RandomForest and VGG16 + LightGBM (92.02%).

Figure 12 shows the accuracy of each model combination for blue geometries. Blue squares: VGG19 + Catboost and VGG19 + AdaBoost had the highest accuracy (97.96%), whereas EfficientNetV2L had the lowest accuracy when combined with ensemble learning. Blue circles: VGG16 + GradientBoosting had the highest accuracy (98.31%), followed by VGG16 + LightGBM, VGG16 + XGboost, and InceptionV3 + Catboost, which were tied for second place (96.61%). Blue diamonds: VGG16 + RandomForest had the highest accuracy (96.15%), followed by VGG16 + LightGBM and VGG16 + XGboost (both 94.23%). EfficientNetV2L had the lowest accuracies with ensemble learning. Blue ovals: VGG19 + Catboost had the highest accuracy (98.18%), whereas EfficientNetB0 had the lowest accuracies with ensemble learning. Blue stars: InceptionV3 + Catboost had the highest accuracy (96.36%). Blue triangles: EfficientNetB0 + Catboost had the highest accuracy (96.23%), followed by EfficientNetB0 + GradientBoosting and ResNet50 + RandomForest, which both had an accuracy of 90.57%. All blue geometries: VGG16 + RandomForest had the highest overall accuracy (93.73%), followed by VGG16 + LightGBM (92.74%). EfficientNetV2L had the lowest accuracies with ensemble learning.

## 4. Discussion

According to the literature, bagging ensembles are outperformed by boosting ensembles in most scenarios [71]. However, our experimental results indicated that this does not hold true for surface defect detection in 3D-printed geometries. As shown in Figure 10, Figure 11 and Figure 12, bagging ensembles yielded the highest accuracies. This may be because bagging reduces variance instead of bias, which can prevent overfitting. Furthermore, in contrast to bagging ensembles, the trees in boosting ensembles are correlated with each other. This makes it possible to form incorrect correlations, leading to worse performance compared with bagging.

The performance of a CNN can be significantly improved by increasing its depth or using novel structures, e.g., by using residual learning, factorization, and modules instead of layers, which increases the accuracy [72,73]. However, the results in Figure 10, Figure 11 and Figure 12 indicate that this is not always true. Although deeper networks and novel CNN structures allow for the extraction of finer details, if the details are too fine, the network can be misled. Correct predictions may then be misjudged as being incorrect and vice versa. This phenomenon was apparent in the visualized feature maps of the last layer in the VGG16 and EfficientNetV2L networks (Figure 13), where VGG 16 features 16 layers in the network and EfficientNetV2L features 1028 layers. The last layer of the VGG16 network always contained a well-defined feature, in contrast to EfficientNetV2L, which did not contain a clear feature in its last layer because of its excessive depth, causing the degradation in classification accuracy.

In theory, different filament colors result in different levels of hue saturation and brightness, which can yield significant differences in accuracy. Studies on the correlation between color data and image classification have revealed that the classification accuracy tends to be higher with bright colors [74]. However, the results of this study indicated that the classification accuracy does not differ significantly among the gray-, green-, and blue-colored geometries. This is because the surface defect (roughness) detection is only affected by the surface smoothness and voids (whether the surface is fully filled); as long as the pictures are sufficiently clear, the classification accuracy does not vary significantly with respect to the color.

In industrial manufacturing, ensemble learning can be applied to various sensor data to accurately diagnose and predict faults. Studies [75,76,77,78,79,80] have indicated that ensemble learning can be used to predict and evaluate the performance of industrial machinery and detect faults. In the present study, the mean accuracy of surface quality classification was >90%, and an accuracy of 100% was achieved in some cases. Therefore, it is feasible to evaluate the quality of FDM-printed products by using image recognition technology in conjunction with a CNN.

## 5. Conclusions

Transfer learning with pretrained models was combined with ensemble learning to classify the quality of 3D-printed objects. The objective was to identify the combination of algorithms that yields the highest classification accuracy with variations in object geometry and color. The following conclusions are drawn.

The surface quality of FDM 3D-printed objects can be accurately classified by combining transfer learning with ensemble learning.The combination of VGG16 or VGG19 with ensemble learning gave the highest accuracy for gray-colored geometries. Although model combinations with EfficientNetB0 and EfficientNetV2L exhibited the highest accuracy in a few instances, these models were relatively inaccurate in most situations.Although boosting ensembles usually outperform bagging ensembles, in this case (quality inspection of 3D-printed objects), the combination of a transfer learning model with a bagging ensemble often resulted in better accuracy. Therefore, it was unable to prove that boosting is superior to bagging (or vice versa) in this study.Although deeper networks with novel structures often achieve better CNN performance (and a higher classification accuracy), this rule does not apply to quality inspections for FDM-printed objects.In this study, the highest classification accuracy of the model combinations did not vary significantly with respect to the color and geometry. Therefore, the filament color does not significantly affect the classification accuracy.

In our future work, we will develop real-time solutions to monitor 3D printing quality and detect printing failures by combining machine learning with a camera module. In addition, we will continue to incorporate the latest machine learning techniques for increasing the overall classification accuracy, to automate visual anomaly detection and eliminate the time and financial costs associated with manual inspections.

## Figures and Tables

**Figure 1 sensors-23-00491-f001:**
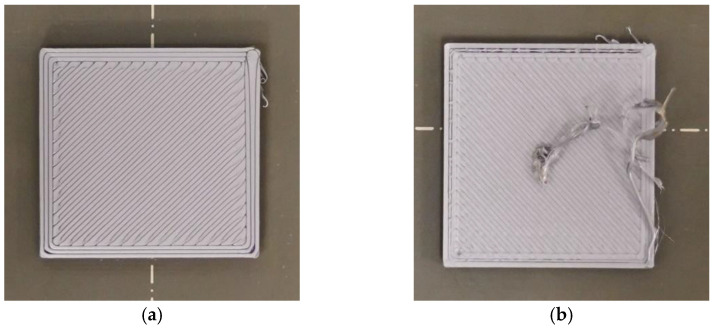
Photographs of (**a**) nondefective and (**b**) defective samples.

**Figure 2 sensors-23-00491-f002:**
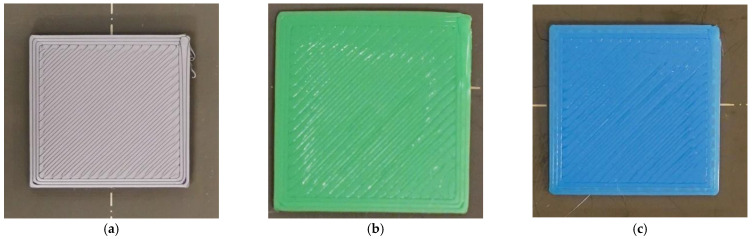
Photographs of (**a**) gray, (**b**) green, and (**c**) blue samples.

**Figure 3 sensors-23-00491-f003:**
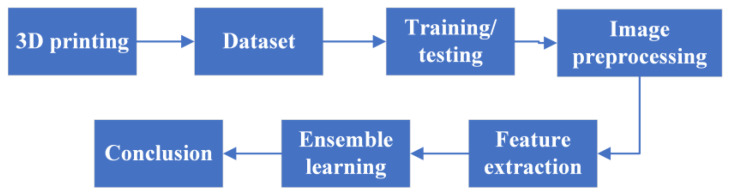
Experimental procedure.

**Figure 4 sensors-23-00491-f004:**
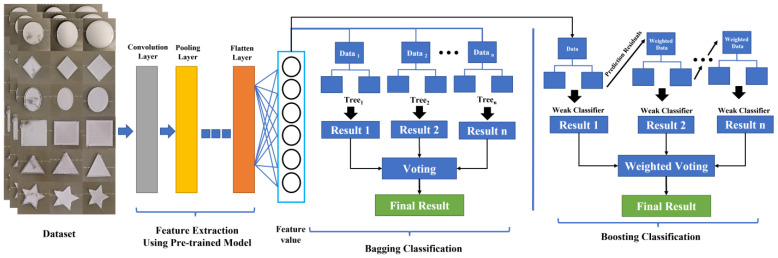
The flow graph of the proposed method for bagging or boosting classification.

**Figure 5 sensors-23-00491-f005:**
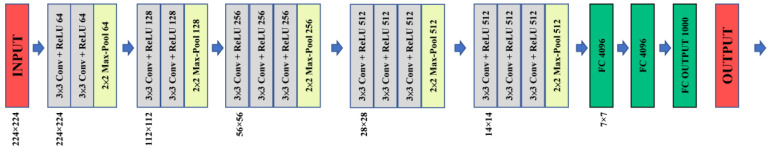
VGG16 architecture.

**Figure 6 sensors-23-00491-f006:**
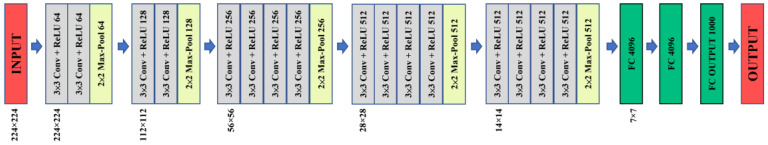
VGG19 architecture.

**Figure 7 sensors-23-00491-f007:**
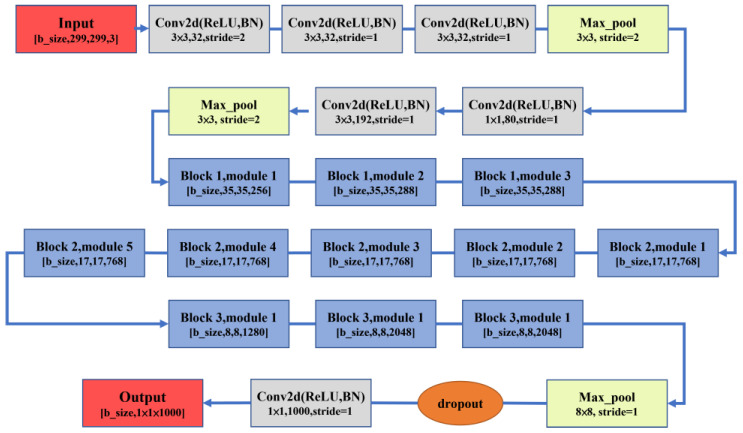
InceptionV3 architecture.

**Figure 8 sensors-23-00491-f008:**
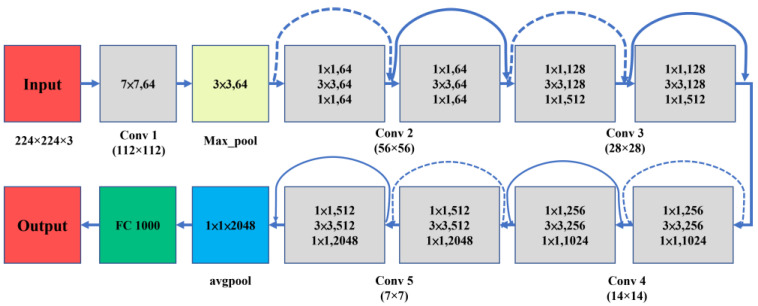
ResNet50 architecture.

**Figure 9 sensors-23-00491-f009:**
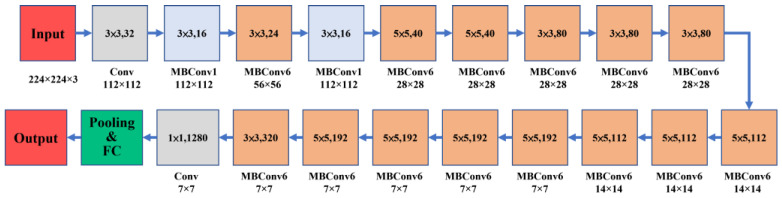
EfficientNetB0 architecture.

**Figure 10 sensors-23-00491-f010:**
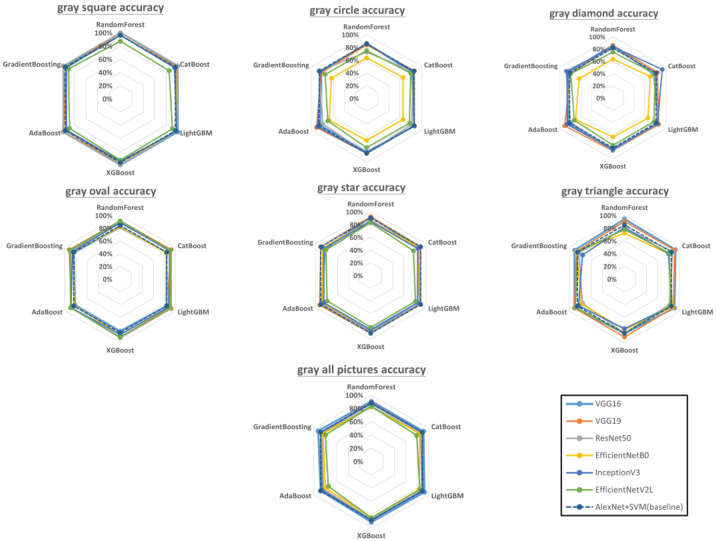
Accuracy of each model combination for gray-colored geometries (raw data are presented in Appendix A). The dotted regular hexagon is the baseline using AlexNet with SVM.

**Figure 11 sensors-23-00491-f011:**
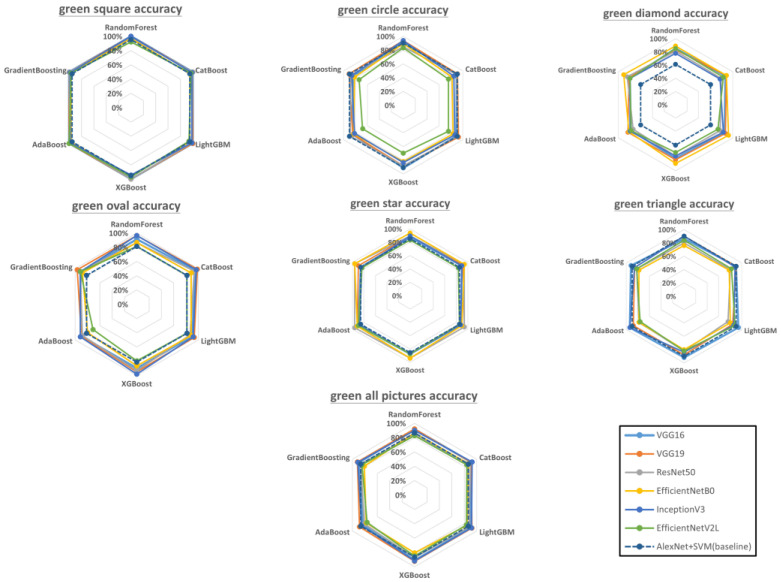
Accuracy of each model combination for green-colored geometries (raw data are presented in Appendix B). The dotted regular hexagon is the baseline using AlexNet with SVM.

**Figure 12 sensors-23-00491-f012:**
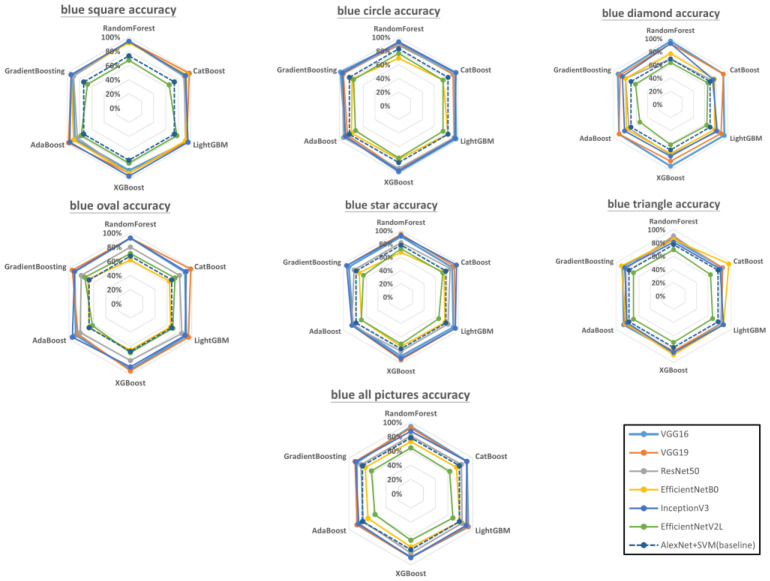
Accuracy of each model combination for blue-colored geometries (raw data are presented in Appendix C). The dotted regular hexagon is the baseline using AlexNet with SVM.

**Figure 13 sensors-23-00491-f013:**
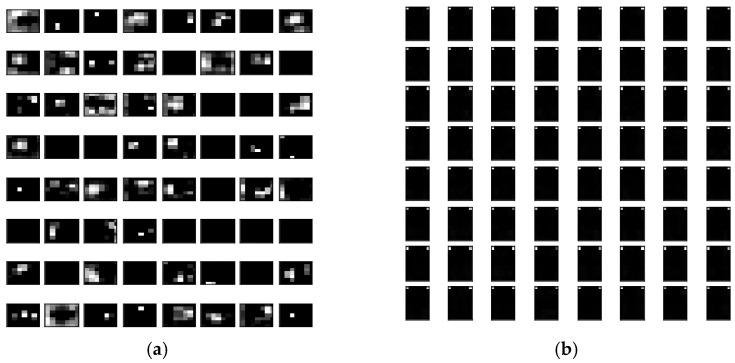
Visualized feature maps of the last layer in the (**a**) VGG16 and (**b**) EfficientNetV2L networks.

**Table 1 sensors-23-00491-t001:** EfficientNetV2-L architecture.

Stage	Operator	Stride	Channels	Layers
0	Conv3 × 3	2	32	1
1	Fused-MBConv1, k3 × 3	1	32	4
2	Fused-MBConv4, k3 × 3	2	64	7
3	Fused-MBConv4, k3 × 3	2	96	7
4	MBConv4, k3 × 3, SE0.25	2	192	10
5	MBConv6, k3 × 3, SE0.25	1	224	19
6	MBConv6, k3 × 3, SE0.25	2	384	25
7	MBConv6, k3 × 3, SE0.25	1	640	7
8	Conv1 × 1 & Pooling & FC	-	1280	1

**Table 2 sensors-23-00491-t002:** Statistics of the dataset.

	Gray Nondefective	Gray Defective	Green Nondefective	Green Defective	Blue Nondefective	Blue Defective
Square	90	104	144	114	117	83
Star	124	134	97	119	76	90
Circle	112	98	156	126	112	127
Oval	71	122	107	131	124	107
Diamond	62	106	134	133	108	105
Triangle	116	136	97	106	107	118

## Data Availability

The data presented in this study are available on request from the corresponding author.

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
