# Peer review of "Three-Dimensional Printing Quality Inspection Based on Transfer Learning with Convolutional Neural Networks"

_sensors, 2023, doi:10.3390/s23010491_

Round 1

Reviewer 1 Report

The authors utilized transfer learning to categorize the quality of items manufactured by FDM-based three-dimensional printing for industrial applications.

While the technique itself is commendable, the novelty of the study is not fully emphasized. Because the study utilized only state-of-art models. The article provides no evidence of innovation in the procedures utilized in the study. The article is well-written, yet it lacks enough innovation. It is suggested that novality be strengthened and/or emphasized.

I would also like to point out the following few points:

Only state-of-the-art models have been applied to this subject. Novality should be increased.

The Introduction section should be expanded to include similar studies.

There is no mention of the data set's properties. How many photos of each size and category does the dataset include?The gray square, gray circle, gray diamond, etc. in Figure 9. What do their definitions refer to? The dataset is not well explained, making it difficult to understand.

Author Contributions section should be edited.

Author Response

Point 1: "While the technique itself is commendable, the novelty of the study is not fully emphasized. 
Because the study utilized only state-of-art models. 
The article provides no evidence of innovation in the procedures utilized in the study. 
The article is well-written, yet it lacks enough innovation. 
It is suggested that novality be strengthened and/or emphasized."
"Only state-of-the-art models have been applied to this subject. Novality should be increased."

Response 1: Thanks for the comment. We have added more descriptions to clarify the contribution of our work, as shown in the 2nd to 5th paragraph in p.3.

Point 2: "The Introduction section should be expanded to include similar studies."

Response 2: Thanks for the suggestion. We have expanded the discussion to include more related works, as shown in the 4th and 5th paragraph of p.2.

Point 3: "There is no mention of the data set's properties. How many photos of each size and category does the dataset include?The gray square, gray circle, gray diamond, etc. in Figure 9. What do their definitions refer to? The dataset is not well explained, making it difficult to understand."

Response 3: Thanks for pointing this out. We have added dataset descriptions at the beginning of Sec. 3.

Point 4: "Author Contributions section should be edited."

Response 4: We have edited the Author Contributions section, which has followed the format specified by the Sensors journal.

Reviewer 2 Report

In this paper, transfer learning is combined with ensemble learning for 3D print quality classification. Experimental results show that VGG models have the best performance among all the used pre-trained models. More comments are listed as follows:

1. The contributions can be summarized at the end of the introduction. And the proposed models are all pre-trained ones that lack novelty.

2. The network architectures are all well-known. It would be better to provide the general network of the proposed method and specific modifications rather than the existing networks.

3. In the experiments, except for the proposed models, comparisons with recent state-of-the-arts should be provided.

4. Apart from the 3D print quality classification, deep learning-based image quality assessment is very relevant to this task, where some works are recommended to be reviewed:  dual-stream interactive networks for no-reference stereoscopic image quality assessment (3D), MC360IQA a multi-channel CNN for blind 360-degree image quality assessment (VR), etc.

5. Please further improve the presentation: Some of the experimental settings lack explanations, e.g. the used datasets.

Author Response

Point 1: The contributions can be summarized at the end of the introduction. And the proposed models are all pre-trained ones that lack novelty.

Response 1: Thanks for the comment. We have added more descriptions to clarify the contribution of our work, as shown in the 2nd to 5th paragraph in p.3

Point 2:  The network architectures are all well-known. It would be better to provide the general network of the proposed method and specific modifications rather than the existing networks.

Response 2: Thanks for the suggestion. We have added a flow graph of our proposed method as shown in Figure 4.

Point 3: In the experiments, except for the proposed models, comparisons with recent state-of-the-arts should be provided.

Response 3: We have added the experimental results of using the combination of AlexNet with SVM as the baseline for comparison. The related work [13], which also utilized pretrained models for feature extraction, reported that AlexNet with SVM achieves the best accuracy. The accuracy of AlexNet with SVM is plotted as dotted regular hexagons in Figure 10, 11, and 12.

Point 4: Apart from the 3D print quality classification, deep learning-based image quality assessment is very relevant to this task, where some works are recommended to be reviewed:  dual-stream interactive networks for no-reference stereoscopic image quality assessment (3D), MC360IQA a multi-channel CNN for blind 360-degree image quality assessment (VR), etc.

Response 4: We have extended the discussions of related works to include the image quality assessments, as shown in the first paragraph and the end of 2nd paragraph in p.2

Point 5: Please further improve the presentation: Some of the experimental settings lack explanations, e.g. the used datasets.

Response 5: We have added dataset descriptions as well as the 3D printing parameters at the beginning of Sec. 3.

Reviewer 3 Report

The paper is very interesting and will be a good contribution towards the research in 3D printing.

I have several comments.

1. Can you elaborate the different results obtain in the gray colored geometry. Explain why different method provides better result compare to the others. 

2. In your discussion, please elaborate your statement in "Correct predictions may then be misjudged as being incorrect and vice versa".  How can the correct prediction being made .

3. Based on your statement colors does not have significant changes, however did you compare the changes between white and black filaments?

4. Can you clarify the several parameters which will help other researchers into better understand the paper.

(a) printing temperature (extruder and bed)

(b) Fill density

(c) Retraction

(d) Print speed

(e) Layer thickness

(f) top solid layers

5. Based on the Figure 2. Is the structure finish printing of just half way. It looks like the structure is printing the infill not the finished part.

Author Response

Point 1: Can you elaborate the different results obtain in the gray colored geometry. Explain why different method provides better result compare to the others. 

Response 1: Thanks for the comment. The extracted features are different in every color-shape combination and different bagging and boosting methods are utilized, which may cause different results in classification accuracy. Therefore, our objective is finding the combination with best accuracy for each color-shape.

Point 2: In your discussion, please elaborate your statement in "Correct predictions may then be misjudged as being incorrect and vice versa".  How can the correct prediction being made .

Response 2: We have expanded the discussion in the 2nd paragraph of Sec. 4 to clarify this:" Although deeper networks and novel CNN structures allow the extraction of finer details, if the details are too fine, the network can be misled. Correct predictions may then be misjudged as being incorrect and vice versa. This phenomenon was apparent in the visualized feature maps of the last layer in the VGG16 and EfficientNetV2L networks (Figure 13), where VGG 16 features 16 layers in the network and EfficientNetV2L features 1028 layers. The last layer of the VGG16 network always contained a well-defined feature, in contrast to EfficientNetV2L, which did not contain a clear feature in its last layer, because of its excessive depth, causing the degradation in classification accuracy."

Point 3: Based on your statement colors does not have significant changes, however did you compare the changes between white and black filaments?

Response 3: Thanks for the suggestion. This work is currently limited on the investigation of the differences in classification accuracy between bright colors. We will extend our samples to include white and black filaments in the future work.

Point 4: Can you clarify the several parameters which will help other researchers into better understand the paper.
(a) printing temperature (extruder and bed)
(b) Fill density
(c) Retraction
(d) Print speed
(e) Layer thickness
(f) top solid layers

Response 4: We have added the descriptions of the parameters at the beginning of Sec. 3.

Point 5: Based on the Figure 2. Is the structure finish printing of just half way. It looks like the structure is printing the infill not the finished part.

Response 5: Figure 2 is the picture of the finished layer.  To clarify this, we have added the following descriptions in the first paragraph of Sec 2.1: "Figure 2 shows the pictures of the finished layer. The collected dataset contains the pictures of every printing layer."

Round 2

Reviewer 1 Report

I thank the authors for taking my suggestions into consideration. When the revised version of the article is examined, it is seen that there is an improvement.

My only minor suggestion is this:

It would be more appropriate to use the × symbol instead of * in Figures 5, 6, 7, 8 and 9.

Author Response

Point 1: It would be more appropriate to use the × symbol instead of * in Figures 5, 6, 7, 8 and 9.

Response 1: We have made changes accordingly in Figure 5 to 9.

Reviewer 2 Report

The authors have addressed most of my comments. There are two final results in Fig. 4. And the typeface of figures should be unified, e.g., Fig. 3 and Fig. 4.

Author Response

Point 1: There are two final results in Fig. 4. 

Response 1: In the proposed method, either bagging or boosting will be used in the classification task. To clarify this, we have revised the caption of Figure 4 to
"The flow graph of the proposed method for bagging or boosting classification."

Point 2: And the typeface of figures should be unified, e.g., Fig. 3 and Fig. 4.

Response 2: Thanks for pointing this out. The typeface in Figure 3 and 4 has been unified.